# Nighttime Walking with Music: Does Music Mediate the Influence of Personal Distress on Perceived Safety?

**DOI:** 10.3390/ijerph19031383

**Published:** 2022-01-26

**Authors:** Ga Eul Yoo, Sung Jin Hong, Hyun Ju Chong

**Affiliations:** 1Department of Music Therapy, Graduate School, Ewha Womans University, Seoul 03760, Korea; bbird27@ewha.ac.kr; 2KU Program in Urban Regeneration, Graduate School, Korea University, Seoul 02841, Korea; yajinhong@gmail.com; 3Strategic Sales HQ Consulting Group, S-1 Corporation, Seoul 04511, Korea

**Keywords:** nighttime walking, environmental music, milieu therapy, female university students

## Abstract

There is growing interest in identifying the environmental factors that contribute to individuals’ perceptions of safety and sense of well-being in public spaces. As such, this study examined how music listening during nighttime walking influenced female university students’ psychological state and perceptions of their campus. A total of 178 female university students with a mean age of 23.0 years participated in this study. One group of 78 students listened to prerecorded music while walking across their campus at night, while the other 100 students did not listen to music during nighttime walking. Immediately following their nighttime walking, participants were asked to rate their psychological state, perceptions on the safety of their campus, and the music (only for the music-listening group). For the non-music-listening group, significant correlations were found between the perceived safety of the campus and psychological states (both anxiety and psychological distress); the correlations were not significant in the music-listening group. The results indicate that music can mediate psychological states, supporting the proactive use of music as a psychological resource for coping with their perceptions of adverse environments. Given the limitations of this preliminary study, further studies with controlled music listening conditions, type of music, and environmental issues are suggested.

## 1. Introduction

Research on music-induced changes in the psychological state of listeners supports the use of environmental music, including background music, to influence people’s mood and behavior [1]. For example, individuals who listened to music embedded in a public place were more likely to judge that space as positive and comfortable [2], and background music in hospital waiting rooms, surgical wards, and intensive care units was found to alleviate patients’ perioperative distress (e.g., anxiety and psychological pain) [3,4]. While environmental music can contribute to a pleasant atmosphere and maintain or elicit desired emotional responses from users of those spaces [5], it can also impact physiological states [6]. For instance, background music has been used in medical settings to stabilize patients’ physiological responses (e.g., heartbeat, blood pressure, respiratory rate, and cortisol levels) that impact recovery from illness [7,8]. Specific attributes of background music (e.g., musical elements that induce positive mood or relaxation and music preference) have been reported to divert listeners’ attention away from the unpleasant features of their internal and external environment and to regulate listeners’ emotions [4].

The research on environmental music is still developing, and recent efforts are focused on expanding its uses. One emerging area of study involves the use of environmental music to deter criminal or antisocial behavior and to facilitate users’ perceptions of safety [9]. The use of music in this context is related to crime prevention through environmental design (CPTED). CPTED involves not only the design of physical security (e.g., barriers and surveillance) but also the enhancement of users’ positive perceptions of users of a space [10,11]. As part of CPTED, environmental music could potentially impact listeners’ perceptions of their environment and induce desirable emotional responses. This is important because when individuals feel secure and stable, they are more likely to take part in preventive activities that create a defensible space [11].

Although the use of music in this context is new, supportive evidence is emerging. In previous studies [9,12], the incorporation of vocal sounds or instrumental music into a public car park or tunnel increased users’ perceptions of the area as safe and resulted in increased use of the spaces. In another study, the number of strangers who loitered around an area at night varied depending on the genre of music played via a speaker-embedded streetlight in a high-crime area [13]. The authors reported that fewer people loitered around the streetlight when the speaker played classical music, which was described as more predictable and less variable in terms of its structure than popular music. This suggests that music can be harnessed to influence listeners’ perception of a specific area, which resulted in decreases in anti-social behaviors, such as loitering behaviors [9]. These results also suggest that different types of music differentially affect users’ perceptions of an area, including their sense of ownership or belonging in that space. This may affect a person’s decision to stay or leave an area. Despite the weak causality between music and criminal activity, there is a growing body of research indicating that people’s perceptions of and behavior in a space can be influenced by environmental music.

The effectiveness of CPTED on university campuses has emerged as an area of significant interest [14,15,16]. CPTED on university campuses focuses not only on decreasing criminal activity but also on positively influencing students’ perceptions of and engagement in their campus environment. Previous studies showed that students who feel safer or more stable are better able to be alert to their surroundings and respond appropriately if there is an issue [17]. Furthermore, students who perceive their campus as safe and feel attached to their campus are more actively engaged in their learning environment [18,19]. These research findings give rise to the need for initiatives to influence students’ perceptions of their campus. As such, environmental music might play a role in helping students positively assess their campus environment and regulate any negative affect associated with their campus. In Korea, there has been a push to create university campuses where students feel safer and less anxious, particularly during nighttime [18,19]. Much of the research in this area presents useful guidelines for designing the physical aspects of safer campus environments.

However, in order to apply the previous finding to situations in Korea, we need to consider cultural aspects [18], as the general learning environment, psychological issues of students, crime rate, and the safety of the campus are different between Korea and western countries. As a preliminary study, this research examined how music listening impacted female students’ perceptions of their university campus. As such, this study aimed to investigate whether the use of music could be an effective component of environmental design of university campuses. This study focused on female university students because research has shown that female students experience significantly higher levels of fear and anxiety on campus than male students [20,21]. Therefore, the purpose of this study was to examine the impact of music during nighttime campus walks on female students’ perceptions of their campus environment and their psychological state.

## 2. Materials and Methods

### 2.1. Research Design

This study was a descriptive study that collected participants’ responses using an online survey. Each participant completed a single survey and the data collected were analyzed to examine differences in perceptions between subgroups and the correlation between the primary variables in this study.

### 2.2. Participants

All procedures and ethical issues related to this study were reviewed and approved by the Institutional Review Board of Ewha Womans University (IRB No. ewha-202004-0007-01). The participants were female undergraduate and graduate students between 20 and 29 years of age who walked on campus at night within the study period. In this study, nighttime was defined as 6 p.m. to 6 a.m. This time period (i.e., after sunset) was based on previous studies in this area and reflects people’s increased fear of crime after sunset [20,22]. A flyer describing the study’s purpose, the research procedures, and the participants’ rights was posted around a university for women located in Seoul, Korea. Each flyer contained one of two QR codes. One code took participants to a web site posting music files and the survey for the music-listening group, and the other code took participants just to the survey for the non-music-listening group. Participants whose QR code was linked to music, were asked to listen to the posted music during their next nighttime walk on campus and then answer the survey questions. Participants whose QR code was not linked to music were asked to respond to the survey questions after their next nighttime walk on campus. All individuals who voluntarily accessed the survey link on the flyer were required to click a button indicating that they agreed to participate in this study. Only after obtaining this consent were participants able to begin the survey. Initially, 204 surveys were submitted, but after removing surveys with incomplete answers, 178 responses (87.3%) were included in the final analysis.

### 2.3. Music

Music for reducing anxiety and inducing a state of relaxation was selected based on previous research findings. Electronic databases were searched using the following keywords: music, music listening, induced emotions, relaxation, anxiety, and psychological distress. The selection criteria for the music included the following: The music had to be observed in an experimental study to change the emotions of listeners and had to be identified as inducing relaxation or reducing psychological distress (e.g., anxiety). In addition, the music could not be accompanied by lyrics. Nine studies met these criteria [23,24,25,26,27,28,29,30,31] and were reviewed in full. The specific musical attributes of these studies were analyzed: the tempo of the music ranged from 50–90 bpm, and there was little variability of form, tempo, or loudness. Given that the participants were university students who were likely exposed to diverse genres of music, the selected music was not limited to classical music, which is the main focus of the music and emotion literature. Accordingly, film music was included. The final 10 selections are displayed in Table 1. Music selections were uploaded through YouTube, and only the subgroup of participants whose QR code was linked to music was able to access the music.

### 2.4. Procedures

The survey was available for four weeks from November 2020 to December 2020. Group assignment was randomly determined and depended on the QR code scanned by the participants. During weeks 1 and 3, the link only took the participants to the survey, which asked the participants to complete the questions immediately after their next nighttime walk across campus. During weeks 2 and 4, the link took the participants to the uploaded music and survey, which asked participants to listen to the music via the link during their next nighttime walk on campus and then to immediately complete the survey. As a result, participants were divided into two groups: One group of participants listened to the selected music while walking on campus at nighttime, and the second group did not listen to the selected music while walking on campus at nighttime. Both groups were asked to complete the participant survey immediately after their next nighttime walking across the campus. The outline of procedures is displayed in Figure 1.

### 2.5. Survey

The survey consisted of three parts: four items related to demographic information (i.e., sex, age, year in school, and duration of university attendance), three items related to the participants’ nighttime walking experience (i.e., time the participant was walking on campus, purpose of walking at that time, and how many people were walking with the participant), and nine items related to their perceptions of their nighttime walk (two items on the perception of safety in general and satisfaction with night walking and seven items on the perception of the walking environment (i.e., pedestrian pathway), such as brightness and tidiness, which have been reported to influence perceived safety [10,11]). Perceptions of safety during nighttime walking were rated using a 5-point Likert scale (1 = “Strongly disagree” to 5 = “Strongly agree”). In addition, participants were instructed to rate their level of anxiety and psychological distress within the past week by selecting a number between 0 (not at all anxious or not at all distressed) and 100 (extremely anxious or extremely distressed).

For the group who listened to music during their nighttime walk, additional survey items asked about their perceptions of the music and how long they listened to the music while walking on campus at night. In addition, there were two questions for each of the following four categories: attentiveness to the music, satisfaction with the music, perceived benefits of the music, and perceived interference of the music. The same 5-point Likert scale that was used to answer questions related to perceptions of campus safety was used to answer these questions pertaining to the music selections. Items related to the perceptions of nighttime walking and perceptions of music are displayed in Appendix A.

### 2.6. Data Analysis

Participants’ survey responses were collected and analyzed using the SPSS Statistics Software 27 (IBM). Given that there were seven survey items on the perceptions of the walking environment, a principal component analysis with an Oblimin rotation was conducted to reduce the dimensions of the dataset and extract the combined features of variables while ensuring further analyses were not biased by multiple comparisons. The component scores extracted from the analysis were used for further analysis.

To examine whether listening to music and listeners’ perceptions of music influences their perceived safety of nighttime walks on their university campus, two subgroups (i.e., participants who listened to music and those who did not listen to music during nighttime walks) were compared. Given that participants’ psychological states might influence their perceived safety, an independent *t*-test measured the differences between the two subgroups in terms of perceived anxiety, psychological distress, and ratings of perceived campus safety. Then, for each group, the measures that were significantly correlated with each component extracted from the principal component analysis were analyzed using a Pearson’s correlation. Age, duration of university attendance, number of walking companions, anxiety level, and psychological distress were analyzed in terms of their correlation to the perceived safety ratings for each group. For the music-listening group, perceptions of the music were also included in the analysis. Furthermore, for the music-listening group, the correlations between the perceptions of music (in terms of benefits and interference), attentiveness to music and satisfaction with music were also investigated.

## 3. Results

### 3.1. Perceived Safety of Nighttime Walking

Initially, 204 surveys were submitted. After reviewing the responses, 26 surveys were excluded from analysis. For example, if the time of survey completion was not within the permitted time frame (6 p.m.–6 a.m.), it was assumed the participant did not complete the survey immediately after completing their nighttime walk and their survey was excluded from the analysis. In addition, if a participant reported that they listened to music for less than 5 min, their survey was not included in the analysis, since not enough time may have passed for them to be influenced by the music.

A total of 178 responses were included in the final analysis. Table 2 displays the participants’ demographic information and the general characteristics of their nighttime walks. The average age of the participants was 23.0 years (SD = 3.5). First-year students made up 13.4% of the participants, while 19.6% were sophomores, 14.5% were juniors, 27.9% were seniors, and 24.6% were graduate students. Most of the participants (72.6%) walked on campus between 6–8 p.m., followed by 10 p.m.–12 a.m. (22.9%). Almost 70% of participants reported that they walked on campus to return to their home/dormitory, and 63.1% of respondents said that they walked alone.

Participants’ perceptions of nighttime walking are displayed in Table 3. The average ratings for safety and satisfaction were 3.6 (*SD* = 0.7) and 3.6 (*SD* = 0.9), respectively. When the reverse coding was reflected and all ratings were analyzed in terms of positive perceptions of a safe environment, the number of ratings for “disagree” and “strongly disagree” corresponded to 6.2% (for safety) and 10.2% (for satisfaction). The average rating for each item is displayed in Table 3. With regard to the perceived level of anxiety and psychological distress felt within the past week, the average rating was 36.2 for anxiety (*SD* = 26.0) and 36.0 for psychological distress (*SD* = 26.2), showing that the average rating was low, given that the maximum marking score was 100.0. When comparing the perceived safety of participants who walked alone with the perceived safety of participants who walked in a group, the average ratings were 3.5 (walking alone; *SD* = 0.7) and 3.7 (walking in a group; *SD* = 0.6). This difference did not reach statistical significance (*t* = −1.662, *p* = 0.098), indicating that the number of companions did not significantly influence the participants’ perceptions of safety in this study.

The results of the principal component analysis, conducted to reduce the dataset for the perception of walking environment (i.e., pedestrian pathway), Bartlett’s test result (χ^2^ = 328.2, df = 21, *p* < 0.001), and the Kaiser–Meyer–Olkin measure of 0.680 (exceeding 0.6) indicated sampling adequacy. The scree plot decided 2 principal components could be extracted, and components 1 and 2 accounted for 60.8% of the total variance. Table 4 displays the survey items and their corresponding factor loading.

For further analysis, the two extracted components were labeled, based on the core concept of CPTED. Then, the items, which loaded on the same component, were averaged and the average ratings were analyzed. The first component with four items loaded was labeled perception on image control/management [11]. In CPTED, image control/management describes the degree to which a certain place is well managed and limits the factors known to attract criminal behavior (e.g., physical obstacles that block users’ sight) and relaxes users or residents. The second component that three items were loaded on was labeled perception on territoriality [11]. Territoriality is related to defensibility and access control, which emphasize the boundary of a certain place and its barriers against intrusion.

The rating for each item that loaded on these two components is displayed in Table 5. The average of the four items that loaded on the first component was higher than the average of the three items that loaded on the second component.

In order to see if these ratings on perceived safety of the campus were influenced by time of night or year in school, a one-way ANOVA was conducted for the average rating. The average rating was not significantly different depending on time of walk, *F*(3, 174) = 0.130, *p* = 0.902 or school year, *F*(4, 173) = 1.541, *p* = 141.

### 3.2. Perception of the Music Listened to during Nighttime Walking

Of the 178 total respondents, 78 (43.8%) listened to music while walking. They reported that they listened to music for an average of 20.7 min (*SD* = 12.3, range of 5 to 60 min). After walking, they were asked to rate the music they listened to in terms of their attentiveness to the music, satisfaction with the music, perceived benefits of music, and perceived interference of music during nighttime walking. The average rating for each item is displayed in Table 6. Respondents tended to report that they were attentive to the music (average rating for the relevant two items = 3.61, *SD* = 0.85), that they were satisfied with the music provided (*M* = 3.88, *SD* = 0.77), that they believed their walking at night was quite safe (*M* = 3.55, *SD* = 0.90), and that the music did not interfere with their nighttime walk (*M* = 2.26, *SD* = 1.07).

When analyzing whether listening to music and the perceptions of music could be influencing factors for perceived safety on campus at night, correlations of the perceived safety between the two groups (i.e., participants who listened to music and those who did not while nighttime walking on campus) were compared. Given that the initial psychological state, not the activity of listening to music, was a confounding factor, an independent *t*-test was conducted to see if there were significant differences in the participants’ ratings on perceived safety, anxiety, and psychological distress. In addition, using the component scores, the ratings on the perceived safety of the campus were compared. The results showed that the two groups were not significantly different in terms of these measures (see Table 7).

### 3.3. Factors Related to the Perceived Safety of Nighttime Walking on Campus

The next analysis examined what was correlated with the perceived safety of nighttime walking. The correlation results are displayed in Table 8 and Table 9. For the non-music-listening group, the perceived safety of nighttime walking on campus was significantly correlated with anxiety, the perception of image control/management, and the perception of territoriality. In other words, the non-music listening participants who felt more anxious within the past week tended to perceive nighttime walking as less safe. However, they felt safer during nighttime walking when they perceived the environment as being better managed and more under control. In terms of satisfaction with nighttime walking, the correlation with age was negative at a significant level, indicating that older participants tended to perceive their campus as less controlled. Increased satisfaction with nighttime walking was associated with decreases in anxiety and psychological distress. Significantly positive correlations were also observed with ratings of the perception of image control/management and territoriality.

For the music-listening group, the ratings of the perceived benefits and interference from music during nighttime walking were added to the correlation analyses. Similar to the non-music-listening group, significant correlations of perceived safety or satisfaction with nighttime walking were observed with the two components of the perceived walking environment. In addition, a significant correlation was observed with the perceived benefits of music for safe walking (Table 9). Participants who perceived music as helping them to feel safer during their nighttime walk tended to perceive their walking environment as more secure. There were no significant correlations with the other physical or psychological measures.

Lastly, there was a significant correlation between satisfaction with the music and feelings of safety (see Table 10). This indicates that participants who were more satisfied with the music perceived their nighttime walk as safer.

## 4. Discussion

A total of 178 female university students participated in this study, and they were asked to rate how safe they perceived their campus environment while walking at night. The participants were divided into two groups. One group listened to prerecorded music while walking at night and the other group did not listen to music while walking at night. The main analyses in this study identified variables associated with the participants’ perceptions on safer walking at night.

Participants in this study perceived the nighttime walking environment on their campus as moderately safe. Less than 10% of the participants rated their campus as unsafe at night, and the average rating of 3.6 for both perceived safety and satisfaction was slightly above the neutral rating of 3. In relation to the two components extracted from the ratings of perceptions on pedestrian pathway via a principal component analysis, participants perceived that the nighttime environment at their university was clean and well ordered (component 1 of perception of image control/management) and that the pedestrian pathways had little access for strangers (component 2 of perception of territoriality) at a moderate to high level. Given that the survey items used in this study were related to the core concept of CPTED in terms of how a well-managed and controlled environment positively influences users’ perceptions of that environment [10,11], this study supports the need for initiatives that promote a greater awareness of campus safety, particularly campus safety at night. Moreover, the participants’ relatively lower ratings of the perception of territoriality indicate the need for greater consideration of the campus’s physical boundaries and access points, which could lead to students’ greater ownership of the campus and facilitate their involvement in learning and other relevant activities [16,17]. Still, these results related to female students’ perceived campus safety should be generalized with caution. This study included only one university, which was located in a big city with a lower crime rate. Although, as a preliminary study, this study focused on female students in order to control confounding variables and clarify the influences of primary variables in a more homogeneous group, the fact that participants were enrolled at a women’s university might have influenced their safety ratings, which would limit direct application to other universities. Future studies should include more universities from more diverse areas across Korea. Moreover, the examination of the actual physical environment was not included in this study that focused solely on how university students perceived their campus environment. Inclusion of the physical index will provide a more integrated view for the future environmental design of university campuses.

In terms of the associations between the perceived safety of nighttime walking on campus and participants’ demographics (e.g., age, school year, and months at the university), walking environment (e.g., number of companions), perceptual factors (perception of the walking environment), and psychological factors (e.g., anxiety and psychological distress), when the music-listening and non-music-listening groups were compared, these two groups showed similarities and differences. For both groups, the variables of age, duration at university, and number of companions were not correlated with their perception at a significant level. However, how they perceived their walking environment (i.e., perception of image control/management and territoriality) was positively correlated with perceived safety for both groups at a significant level. The finding that the number of people accompanying a participant on their nighttime walk was not as influential in determining their feelings of safety as their perception of the space being free of strangers gives rise to the need to consider the perceived environment as well as physical environment for CPTED. This is especially important for university campuses, where the perception of crime may not accurately reflect crime statistics. This aligns with the discussion of defensible space from the CPTED perspective [11], which applies to the general community. This concept emphasizes people’s (users or residents) perceptions of a space as being an agent for influencing their reactions to that space [10]. One such perception of a space involves ownership, which has been highlighted in multiple studies [11], and for this perception, various strategies, including supportive activities, were suggested to encourage safe activity and thereby discourage illegitimate or criminal activity. As such, the efforts to promote supportive activities are increasing [32], and this preliminary study presents implications for how to incorporate music into environmental design for university campuses to influence students’ perceptions of their campus and their psychological state.

An interesting finding from this study lies in the differences between the music listening and non-music-listening groups. While the non-music-listening group showed a significant correlation between perceived safety and anxiety and/or psychological distress, the music-listening group did not exhibit this correlation, although the groups were not significantly different in their level of perceived anxiety or psychological distress. Female university students who did not listen to music while they were walking on pedestrian pathways at night tended to perceive nighttime walking as less safe as their anxiety or psychological distress increased. This is supported by a study showing that people with higher levels of psychological distress feel less secure [33]. Another study [17] found that perceived safety was related to the degree of balance between individual resources and demands. In that context, the ability to utilize one’s own coping strategies (i.e., feelings of self-competence and supportive relationships) in an environment was considered a mediator of the perceived safety of the environment. According to this argument, participants in this study who felt more anxious might also tend to feel more powerless, which could lead to increased feelings of fear. Unlike the non-music-listening group, the music-listening group did not show this association. Furthermore, for the music-listening group, their perceived safety of nighttime walking was found to be significantly correlated with their perceived benefits of music during their nighttime walking.

Although additional studies are needed to identify the direct relationship or factors for this mediation, this study proposes that music can mediate negative psychological states (i.e., anxiety and psychological distress) so that those states do not negatively affect feelings of safety. This is also supported by a significant correlation between the perceived safety and perceived benefits of music during nighttime walking. Perhaps music serves as an active agent in facilitating an individual’s utilization of their psychological resources, regardless of their psychological state. Furthermore, in terms of the potential role of music for increasing feelings of safety during nighttime walking, this study demonstrates that as participants were more attentive to music and more satisfied with the type of music they listened to, they tended to perceive the music as making them feel safer. Depending on how music is perceived, the perceived benefits of music may vary. In this context, more specified ways of using music should be investigated through future studies. This study aligns with previous research supporting the use of music in influencing listeners’ perceptions of a space. This body of work indicates that different types of music have differential effects on regulating the emotions of listeners [9].

Despite the potential benefits of using music to influence students’ perceived safety of their campus at night, there are some limitations related to this study. Given that this study examined personal listening to music, listening time was not controlled and listening time ranged from 5 min to 60 min. Moreover, the study participants were asked to rate their psychological state over the past week, and this may have obfuscated actual pre-post changes after music listening. Although the main focus of this preliminary study was on the applicability of music listening itself for coping with psychological issues related to nighttime walking, the extent to which music impacts people’s perceptions of safety, anxiety, or psychological distress should be further examined in future studies. In addition, given that this study used music that was reported to induce relaxed emotional states, the use of other types of music (e.g., different musical styles or the participants’ preferred or selected music) would present more specific implications for how to select and use music in this area.

In this study, participants used their own device to listen to music (e.g., earbuds). It is important to note that individuals should use caution when listening to music and ensure that they are able to hear other sounds in their environment [34]. Even though participants in this study reported low levels of interference from the music while walking, additional factors for using music for nighttime walking should be identified prior to generalized application of music for such a purpose.

## 5. Conclusions

In Korea, there has been increasing attention paid to CPTED for designing and improving campus environments. Today, CPTED extends beyond the previously emphasized management of a physical space (e.g., surveillance equipment and increases in police officers) to the perceptions of the users of a specific space and their psychological state. Given that attempts to identify diverse options for more integrated and expanded implementation of CPTED are still lacking, this study offers preliminary work on incorporating music listening, which has repeatedly been reported to influence listeners’ psychological state and perception of their environment, into CPTED efforts. Moreover, it supports the potential role of music listening in environmental design in terms of mediating the perceptual and psychological factors affecting the perceived safety of nighttime walking on a university campus. Nevertheless, since this study did not confirm the direct effects of music for reducing anxiety or psychological distress or enhancing perceived safety and only identified the correlation between these measures, further studies are needed to confirm the effects of music in this context. Furthermore, future studies could propose evidence-based ways of selecting and using music to obtain the desired effects.

## Figures and Tables

**Figure 1 ijerph-19-01383-f001:**
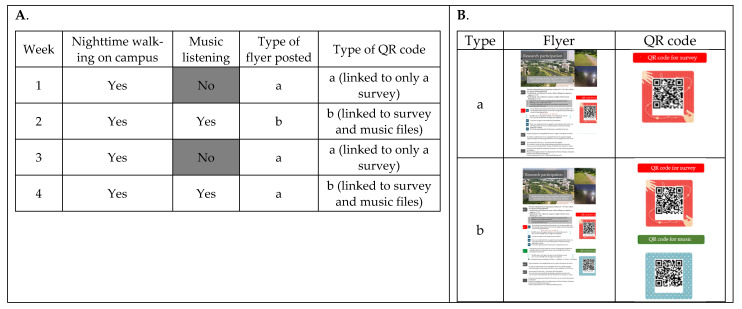
The outline of procedures. Panel (**A**) indicates the condition of nighttime walking on campus and the type of flyers used. Panel (**B**) shows the flyers that were posted and the QR codes that were embedded in the flyer.

**Table 1 ijerph-19-01383-t001:** List of selected music.

Composer	Title of Music	Genre	Instrument	Reference	Tempo (bpm)
E. Elgar	Salut d’Amour	C	Guitar	[23]	72
E. Grieg	Peer Gynt Suite No. 1, Op. 46 “Morning”	C	Flute/Harp	[24]	60
W. A. Mozart	Piano Sonata K.570 2nd movement	C	Piano	[25]	80
J. Pachelbel	Canon	C	Guitar	[26]	52
S. Rachmaninoff	Rhapsody on a Theme of Paganini, Op. 43. Var. 18.	C	Pianos	[27]	79
R. Schumann	Traumerei, Op. 15, No. 7	C	Piano	[28]	84
J. Barry	If I Know a Song of Africa from “Out of Africa”	F	Flute/Strings	[29]	81
D. Marianelli	The Secret Life of Daydreams from “Pride & Prejudice”	F	Piano/Strings	[30]	70
D. Marianelli	Dawn from “Pride & Prejudice”	F	Piano	[30]	66
A. Johnston	Rose Garden from “Becoming Jane”	F	Piano/Strings	[31]	74

Note. bpm: beats per minute; C: classical; F: film music.

**Table 2 ijerph-19-01383-t002:** Participants’ demographic information.

Variable	*N* = 178
Sex, F:M	178:0
Age (years), *M* (*SD*)	23.0 (3.5)
Year in school	
1st year	24 (13.4%)
2nd year	35 (19.6%)
3rd year	26 (14.5%)
4th year	50 (27.9%)
Graduate	44 (24.6%)
Duration of university attendance (months), *M* (*SD*)	32.9 (22.6)
Time period of walking on campus at night	
6–8 p.m.	130 (72.6%)
8–10 p.m.	0 (0.0%)
10 p.m.–12 a.m.	41 (22.9%)
12–2 a.m.	6 (3.4%)
2–6 a.m.	2 (1.1%)
Purpose of walk	
Returning home	124 (69.3%)
Exercising	15 (8.4%)
Walking to a night class	11 (6.1%)
Walking to a campus facility for meeting or event	15 (8.4%)
Walking to the library	13 (7.3%)
Walking to get food	1 (0.5%)
Number of companions	
0 (Walking alone)	113 (63.1%)
1	34 (19.0%)
2	15 (8.4%)
3	8 (4.5%)
4	4 (2.2%)
5	5 (2.8%)

Note. F: female; M: male.

**Table 3 ijerph-19-01383-t003:** Participants’ perceptions of their nighttime walk and their psychological state.

Item	*N* = 178
Perceived safety of their last nighttime walk	3.6 (0.7)
Satisfaction with their last nighttime walk	3.6 (0.9)
Level of anxiety felt within the past week	36.2 (26.0)
Level of psychological distress felt within the past week	36.0 (26.2)

**Table 4 ijerph-19-01383-t004:** Principal component analysis results for the ratings of the walking environment.

Item	Component	
	1	2
The pedestrian pathway was clean.	**0.846**	0.065
The environment was well-organized.	**0.835**	0.043
I was able to see the path clearly.	**0.715**	0.005
The pathway was familiar.	**0.558**	−0.546
The pathway was bright.	−0.060	−**0.765**
I saw many people walking around me.	0.328	−**0.682**
I saw many strangers walking around me. *	0.346	**0.636**

* Reverse coding was used since a higher score indicated a negative outcome (i.e., more strangers). Bolded numbers indicate the highly loaded coefficient on each identified principal component.

**Table 5 ijerph-19-01383-t005:** Participants’ perceptions of the pedestrian pathways on campus that they walked at night.

Item	*N* = 178
Items loaded on perception of image control/management	
The pedestrian pathway was clean.	4.0 (0.7)
The environment was well-organized.	3.8 (0.8)
I was able to see the path clearly.	3.4 (0.8)
The pathway was familiar.	4.0 (0.7)
Average of the four items	3.8 (0.6)
Items loaded on perception of territoriality	
The pathway was bright.	3.2 (0.8)
I saw many strangers walking around me. *	3.1 (0.5)
I saw many people walking around me.	2.6 (1.0)
Average of the three items	3.1 (0.5)

* Reverse coding was used since a higher score indicated a negative outcome (i.e., more strangers).

**Table 6 ijerph-19-01383-t006:** Participants’ perceptions of the music listened to during nighttime walking.

Category	Item	*n* = 78
Attentiveness to music	I was aware of the music being played.	3.72(0.78)
	I was attentive to the music.	3.50(0.91)
Satisfaction with the music provided	I was satisfied with the type of music played.	3.86(0.82)
	I was satisfied with the loudness of the music.	3.91(0.72)
Benefits of the music for feeling safe during nighttime walking	Music made me feel safer while walking.	3.35(0.97)
	Music made nighttime walking more satisfying.	3.76(0.78)
Interference of the music during nighttime walking	Music made nighttime walking uncomfortable.	2.32(1.09)
	Music interfered with walking at nighttime.	2.21(1.06)

**Table 7 ijerph-19-01383-t007:** Differences in psychological states and perceptions between the two participant groups.

Variable	Non-Music Listening (*n* = 100)	Music Listening (*n* = 78)	*T*	*p*
Anxiety level	35.2(25.9)	36.7(25.9)	−0.391	0.696
Psychological distress	37.4(28.3)	34.0(23.5)	0.848	0.398
Perception of image control/management ^a^	0.1 (1.0)	−0.1 (0.9)	1.756	0.081
Perception of territoriality ^a^	−0.1 (0.9)	0.1 (1.1)	−1.264	0.208

^a^ The average ratings of the items that loaded on each component were used for the analyses.

**Table 8 ijerph-19-01383-t008:** Correlation between the perceived safety of campus and other measures for the non-music-listening group (*n* = 100).

Variable	Age	# of Months Attending University	Number of Companions	Anxiety	Psychological Distress	Perception of Image Control/Management ^a^	Perception of Territoriality ^a^
Perceived safety of nighttime walking	−0.050	0.076	0.149	−0.268 **	−0.196	0.621 **	0.481 **
Satisfaction with nighttime walking	−0.284 **	0.092	0.183	−0.501 **	−0.369 *	0.594 **	0.386 **

^a^ The average ratings of the items that loaded on each component were used for these analyses; * *p* < 0.05. ** *p* < 0.01.

**Table 9 ijerph-19-01383-t009:** Correlation between the perceived safety of campus and other measures for the music-listening group (*n* = 78).

Variable	Age	# of Months Attending University	Number of Companions	Anxiety	Psychological Distress	Perception of Image Control/Management ^a^	Perception of Territoriality ^a^	Benefits of Music	Interference of Music
Perceived safety of nighttime walking	−0.171	0.067	0.064	0.041	0.098	0.584 **	0.398 **	0.307 **	0.051
Satisfaction with nighttime walking	−0.196	0.082	0.019	−0.005	0.142	0.597 **	0.357 *	0.472 **	0.008

^a^ The average ratings of the items loaded on each component were used for these analyses; * *p* < 0.05. ** *p* < 0.01.

**Table 10 ijerph-19-01383-t010:** Correlation between the perceptions of the music listened to and the benefits or interference of music listening during nighttime walking (*n* = 78).

Variable	Attentiveness to Music	Satisfaction with Music
Perceived benefits of music for feeling safe during nighttime walking	0.521 **	0.525 **
Interference of music during nighttime walking	0.069	−0.064

** *p* < 0.01.

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
