# Peer review of "Nighttime Walking with Music: Does Music Mediate the Influence of Personal Distress on Perceived Safety?"

_ijerph, 2022, doi:10.3390/ijerph19031383_

Round 1

Reviewer 1 Report

ijerph-1515400

Assessment:

Thank you very much for the opportunity to read such an interesting article.

A total of 178 female university students participated in this study, and they were asked to rate how safe they perceived their campus environment while walking at night. The participants were divided into two groups. One group listened to prerecorded music while walking and the other group did not listen to music. Then the main analyses in this study identified variables associated with perception on safer walking at night for each group.

An interesting finding from this study lies in the differences between the music listening and non-music listening groups. While the non-music listening group showed a significant correlation between perceived safety and anxiety and/or psychological dis-358 tress, the music listening group did not exhibit this correlation, although the groups were not significantly different in their level of perceived anxiety or psychological distress. Female university students who did not listen to music while they were walking on pedestrian pathways at night tended to perceive nighttime walking as less safe as their anxiety or psychological distress increased.

The article is very well structured, with adequate methodology. Results and conclusions consistent with the objectives.

Some small considerations to improve it:

1.- It would be interesting to have more details of the process to get to the list of selected music.

2.- I advise putting the survey used as an annex to this article, due to the importance of the study. 

My assessment is positive, you need to check the references according to the standards.

Kind regards.

Author Response

Response to Reviewer 1 Comments

Point 1: It would be interesting to have more details of the process to get to the list of selected music.

Response 1: We added additional statements of how the list of music for this study was formulated.

“Music for reducing anxiety and inducing a state of relaxation was selected based on previous research findings. Electronic databases were searched using the following keywords: music, music listening, induced emotions, relaxation, anxiety, and psychological distress. Selection criteria for the music included the following: the music had to be observed in an experimental study to change the emotions of listeners and had to be identified as inducing relaxation or reducing psychological distress (e.g., anxiety). In addition, the music could not be accompanied by lyrics. Nine studies met these criteria [23-31] and reviewed in full.”

Point 2: I advise putting the survey used as an annex to this article, due to the importance of the study. 

Response 2: We added the Appendix A that includes survey items related to perceptions on the safety of nighttime walking and perceptions of music used during nighttime walking, which we think will help readers understand this study better.

Point 3: My assessment is positive, you need to check the references according to the standards.

Response 3: As suggested, we rechecked the reference style.

Reviewer 2 Report

Interesting work, however, requires some corrections and answers a few issues:
Please justify why the time slot was chosen as nighttime (i.e., 6 pm to 6 am)?
At what time of the year was the survey conducted?
Did nighttime mean a period when it was dark outside?
Was the survey sent after a one-time walk during the nighttime period?
Did researchers have control over whether women actually listened to certain music while walking? How long was the walk? Did it really take place at the assumed nighttime? After what time was the questionnaire filled after the walk?
Is it possible to compare the safety of people walking alone and in a group of people?
Was the day of the week and factors such as alcohol or medications are taken into account?
130 (72.6%) participants walking on campus 6-8 pm are you sure to say that this is nighttime?
Do you have any other demographic data? for example, from what big city did the respondents come from?
The study conditions are specific as the walks took place on the campus of a female university, which may have an impact on the initial safety rating.

Author Response

Response to Reviewer 2 Comments

Point 1: Please justify why the time slot was chosen as nighttime (i.e., 6 pm to 6 am)?

Response 1: We added the explanation of why we defined nighttime from 6 pm to 6 am as following:

“In this study, nighttime was defined at 6 pm to 6 am. This time period (i.e., after sunset) was based on previous studies in this area and reflects people’s increased fear of crime after sunset [20, 22].”

Point 2: At what time of the year was the survey conducted?

Response 2: Survey was administered from November 2020 to December 2020. And we added the statement in the Methods – Procedures section.

Point 3: Did nighttime mean a period when it was dark outside?

Response 3: We defined the nighttime as “after sunset” based on the relevant literature. We added the explanation of how we define the period of nighttime in the Methods (when we explained the inclusion criteria of participants as individuals who walked on campus during nighttime).

Point 4: Was the survey sent after a one-time walk during the nighttime period?

Response 4: Yes. We added the explanation to clarify the survey procedure as following:

“Both groups were asked to complete the participant survey immediately after their next nighttime walking across the campus.”

Point 5: Did researchers have control over whether women actually listened to certain music while walking?

Response 5: We have to determine whether the participants actually listened to the music while walking only based on the participants’ report on how long they listened to the presented music. However, in order to control the influence of music, we only included the responses of participants who listened to music more than five minutes.

Accordingly, we added the statements in the Results section as following.

“If a participant reported that they listened to music for less than 5 minutes, their survey was not included in the analysis, since not enough time may have passed for them to be influenced by the music.”

Point 6: How long was the walk? Did it really take place at the assumed nighttime? After what time was the questionnaire filled after the walk?

Response 6: Unfortunately, we did not control the duration of walk, since the duration was not considered as influencing factor for perceived safety. However, we asked the participant to report whether they walked on campus during nighttime or not. Based on their report, we assumed that they walked during nighttime as we expected. Also, since this is online questionnaire, we were able to check the time that they’ve submitted their responses. if the time of survey completion was not within the permitted time frame (6 pm – 6 am), it was assumed the participant did not complete the survey immediately after completing their nighttime walk and their survey was excluded from analysis.

Point 7: Is it possible to compare the safety of people walking alone and in a group of people?
Response 7: We compared the safety of people walking alone and in a group of people. And the results of the comparison were added as following.

“When comparing the perceived safety of participants who walked alone with the perceived safety of participants who walked in a group, the average ratings were 3.5 (walking alone; SD = 0.7) and 3.7 (walking in a group; SD = 0.6). This difference did not reach statistical significance (t = -1.662, p = .098), indicating that the number of companions did not significantly influence the participants’ perceptions of safety in this study.”

Point 8: Was the day of the week and factors such as alcohol or medications are taken into account?
Response 8: We did not take such factors into account. This study was implemented online. It may assume that participation in the survey was decided by their conscious and voluntary access to the flyer and willingness to participate. Also, in consideration of cultural aspects in Korea, such factors might not have a big impact in this study.

Point 9: 130 (72.6%) participants walking on campus 6-8 pm are you sure to say that this is nighttime?

Response 9: As we responded to the comments above, the relevant studies have defined the nighttime from 6 pm to 6 am. Additionally, this study was conducted within the campus, a lot of public activities are usually implemented during daytime. Therefore, we decided that it would be appropriate to define the nighttime from 6 pm (i.e., after sunset; and the time without daylight).  

Point 10: Do you have any other demographic data? for example, from what big city did the respondents come from?
Response 10: This study was conducted in Seoul, Korea. We added the statement in the Participant section.

Point 11: The study conditions are specific as the walks took place on the campus of a female university, which may have an impact on the initial safety rating.

Response 11: This study is a preliminary study to see whether inclusion of music impacted college students’ perceived safety while walking on the campus at night. As a preliminary study, there were a lot of variables to be controlled so that our primary interests would be addressed. Accordingly, we only focused on female university students because research has shown that female students experience significantly higher levels of fear and anxiety on campus than male students. However, as suggested, the attempts to include only female students should be carefully considered to interpret the results from this study. We also added the related issues and limitations in the Discussion section.

Reviewer 3 Report

It is a very interesting and important study.

I would suggest to add, that further studies could examine differences between different musical styles and composers within the same musical styles, which are listened during nighttime walk.

Author Response

Response to Reviewer 3 Comments

Point 1: I would suggest to add, that further studies could examine differences between different musical styles and composers within the same musical styles, which are listened during nighttime walk.

Response 1: As suggested, we added suggestions for future studies that could investigate the effect of musical styles as following:

“Also, given that this study used music that was reported to induce relaxed emotional states, the use of other types of music (e.g., different musical styles or participants’ preferred or selected music) would present more specific implications for how to select and use music in this area.”